# Assessing the Applications of Earth Observation Data for Monitoring Artisanal and Small-Scale Gold Mining (ASGM) in Developing Countries

**Abdul-Wadood Moomen** [1]**, Pierre Lacroix** [2,3]**, Antonio Benvenuti** [2]**, Marion Planque** [2]**, Thomas Piller** [2]**, Kenneth Davis** [4]**, Manoela Miranda** [4]**, Elsy Ibrahim** [5] **and Gregory Giuliani** [2,3,*]

[1] School of Mines and Built Environment, University of Energy and Natural Resources, Sunyani P.O. Box 214, Ghana; abdul-wadood.moomen@uenr.edu.gh
[2] GRID-Geneva, Institute for Environmental Sciences, University of Geneva, Bd Carl-Vogt 66, CH-1211 Geneva, Switzerland; pierre.lacroix@unepgrid.ch (P.L.); antonio.benvenuti@unepgrid.ch (A.B.); marion.planque@unige.ch (M.P.); thomas.piller@unepgrid.ch (T.P.)
[3] EnviroSPACE Lab., Institute for Environmental Sciences, University of Geneva, Bd Carl-Vogt 66, CH-1211 Geneva, Switzerland
[4] Secretariat of the Minamata Convention on Mercury, United Nations Environment Programme, Chemin des Anémones 11-13, CH-1219 Geneva, Switzerland; kenneth.davis@un.org (K.D.); manoela.miranda@un.org (M.M.)
[5] Minerals Engineering, Materials & Environment (GeMMe), University of Liege, 4000 Liege, Belgium; elsy.ibrahim@novojy.com
* Correspondence: gregory.giuliani@unige.ch

**Abstract:** This paper discusses opportunities to use remote sensing (RS) technologies in addressing the persistent global challenges related to the artisanal and small-scale gold mining (ASGM) sector. The paper uses a systematic literature review to identify, analyze, and synthesize various uses of RS on the detection and monitoring of ASGM activities across the globe. The study covers the use of spaceborne sensors and available opportunities for data access and processing and emphasizes the important role that freely-available data has played in understanding ASGM activities. It discusses applications and opportunities offered in assessing the geospatial and temporal characteristics of ASGM and its impacts on the surrounding environment. Furthermore, it examines different indicators for the detection of ASGM in the landscape. Finally, technological capabilities described in the study are illustrated with case studies in the Democratic Republic of Congo and in Colombia using cloud computing with the Open Data Cube. The case studies demonstrate the identification and quantification of impacts of ASGM activities on land degradation and water turbidity in remote areas and results are disseminated using the MapX platform. This facilitates policy development, implementation, and evaluation in the ASGM context.

**Keywords:** artisanal and small-scale gold mining; Landsat; Sentinels; Earth observations; sustainable land management; land degradation; land cover change

## 1. Introduction

In the last four decades, relevant discussions on artisanal and small-scale gold mining (ASGM) have become prolific in the agendas of various international and regional development platforms in sub-Saharan Africa, Central and South America, and Southeast Asia [1]. Examples include the United Nations (UN) 1978 landmark conference held under the theme: *The Future of Small-Scale Mining*, the Organisation of African Unity 1980 Mombasa conference on *Strategies for Small-Scale Mining and Mineral Industries*, and a UN Interregional Seminar on *Guidelines for Development of Small and Medium Scale Mining* held in Harare, Zimbabwe, in 1993 [2]. Several guiding principles emanate out of these global

attempts towards monitoring ASGM subsector activities. Examples include the 2002 Ya-oundé Vision on Artisanal and Small-scale Mining, the Africa Mining Vision of February 2009, and the 2018 Mosi-oa-Tunya Declaration on Artisanal and Small-scale Mining, Quarrying, and Development [3–6]. All these guiding principles seek enhanced monitoring techniques for optimizing the critical benefits and mitigating the associated costs of the ASGM subsector on the environment.

ASGM activities lead to the contamination of water bodies and soils, deforestation, and biodiversity loss [7,8]. As many as 19 million ASGM operators in the world use mercury in ore processing, which makes the use of mercury in the subsector a global issue [9]. The entry into force of the Minamata Convention on Mercury in 2017 is a typical example of a renewed global effort to overcome the environmental challenges posed by the ASGM subsector. The UN Environment Programme (UNEP) estimates that ASGM operations alone release over 2000 tons of mercury per annum into the environment [10]. However, the informal nature of most ASGM operations makes it difficult to appraise its total impacts on the economy, the people, and their environment. Besides, most monitoring initiatives on mitigating such impacts focus on land, water, vegetation, and society. However, unlike large-scale mining activities, it can be challenging to obtain reliable information about the location and spatial extent of ASGM activities as they are typically remote, dispersed, and often obscure [6,11]. To address this requires a holistic look at the form, shape, and nature of the variables affected by the ASGM activities in the landscape. This relies on a variety of direct and indirect information provided by a diverse array of tools and techniques. This paper, hereby, demonstrates that RS is one such promising approach for direct and indirect monitoring of the environmental impacts of the ASGM subsector [12].

Examples of the use of RS on ASGM monitoring include the works of Barenblitt, et al. [13]. Their study in Ghana found that between 2014 and 2017, approximately 47,000 ha (± 2218 ha) of vegetation were destroyed by ASGM activities at an average rate of ~2600 ha yr$^{-1}$. The World Database of Protected Areas also used RS tools to map about 700 ha of protected areas that have been disturbed by ASGM activities in Latin America and Africa [13]. Other examples include the works of Bruno, et al. [14], Isidro, et al. [15], Ibrahim, et al. [16], Lobo, et al. [17], de Lucia Lobo, et al. [18], Nyamekye, et al. [19], and Telmer and Stapper [20]. These studies have used both optical spaceborne RS data of high and medium resolution to identify and follow the evolution of artisanal and small-scale mines across space and time. For instance, Forkuor, et al. [21] used annual time-series Sentinel-1 and Sentinel-2 data to map and monitor ASGM activities along major rivers in southwestern Ghana. Such temporal analysis of spaceborne data provides decision-makers with an efficient monitoring system of ASGM operations.

However, the applications of simple but efficient RS techniques for monitoring the spatial dimensions of ASGM activities in sub-Saharan Africa, Central and South America, Southeast Asia, and other developing regions are limited [22]. Although RS has contributed to prospecting for gold deposits in low-vegetated areas [23], its capability to contribute to the detection of the occurrences and monitoring the operations of small-scale gold mining has been under-explored, especially in highly vegetated areas. As proposed in the socio-economic ASGM research methodology of UNITAR [24], countries in which ASGM is practiced should be able to produce data on several aspects, including health and environment, to have a clear understanding of the linked impacts of ASGM on their economy. Despite the fact that applicable RS techniques are considered context specific, it is widely acknowledged that the identification of ASGM activities through RS should be the starting point [25]. This helps in identifying ASGM hotspots, defining sensitive areas where national laws and policies are poorly enforced, and initiating local participation.

Despite the broad availability of Earth observation data and the feasibility of RS techniques, there has been limited overall progress and success on the applications of these technologies for characterizing and monitoring the ASGM subsector activities in developing countries. Thus, this paper seeks to encourage the adoption of RS in policy guidelines, which can provide consistent and effective data, upon which discussions on the

environmental protection, sustainability, and livelihood security of the ASGM subsector can be built in developing countries. To this end, the paper assesses the specific applications of RS in monitoring ASGM activities in developing countries and demonstrates the potential and opportunities in the adoption of RS for guiding and informing sustainable practices in ASGM. The paper further assesses the existing methods used and examines different indicators, including the presence of mercury, water pollution, and land degradation, for the detection of ASGM in the landscape. It identifies the possible algorithms for effectively detecting the presence of ASGM, especially in remote areas, and provides guidance for monitoring the use of mercury as a special indicator of ASGM in an area. The paper tests and implements these algorithms through case studies in Africa and Latin America. Finally, the paper recommends the policy needs for adopting such technologies in developing countries. These countries generally have human capacity challenges with regards to the manipulation of relevant RS software, access to data, data security, and interpretation.

## 2. Materials and Methods

### 2.1. Literature Review

This study presents the role of RS with respect to the detection, mapping, and monitoring of ASGM activities and their effects. The paper reviews literature including academic and scientific publications, reports emanating from numerous initiatives on ASGM, laboratory methods, and technology manuals. Thus, the review is organized as follows: (1) spaceborne sensors utilized for studying ASGM activities; (2) approaches of RS for the detection and mapping of ASGM impacts on land, vegetation, and water; (3) findings obtained from these approaches; (4) techniques for the detection of mercury in ASGM settings; and (5) an illustration of the use of spaceborne data in a cloud-computing environment to detect all of the above in ASGM operation areas.

The literature review focuses on the last three decades. The approach taken for searching the relevant literature [26–28] consisted of a set of keywords used to query different repositories such as scientific libraries (e.g., Science Direct, Web of Knowledge, Google Scholar) and personal databases of the researchers and their research groups. The following list of keywords were used individually and combined with each other for each query: "Remote sensing for detecting small-scale mining", "Satellite for monitoring illegal mining activities", "Detection of informal mining activities", "Impacts of artisanal and small-scale mining activities on water and forest", "Mines detection in forest areas", "Monitoring Artisanal and small mining along water bodies", and "the use of RS in detecting artisanal and small-scale mining". This search produced a comprehensive list of articles. To refine the results, three additional criteria were used: articles should address ASGM activities and should include the use of RS as a key tool or supporting tool; the scope of articles considered were those written in either English, French or Spanish (The literature review conducted in French and Spanish was conducted following the same methodology used the beforementioned keywords translated accordingly. That is to say "Télédétection", "Satellite", "Détection", "Imagerie", "Mines", "Minier", "Minière", "Or", "Aurifère", "Extractif", "Extractive", "Extraction", "Artisanal", "Artisanale", "mercure", "Minamata" et "Orpaillage" in French, and "MAPE", "Minería de oro artesanal y de pequeña escala", "EVOA", "explotación de oro de aluvión", "mina", "minería ", "mina artisanale", "minería artisanale", "mina de oro", "minería de oro", "sensores remotos", "imágenes satelitales", "teledetección", "mercurio", and "Minamata" in Spanish); the first 50 records were screened within online scientific libraries to identify the most relevant publications addressing applications of RS for the monitoring and assessment of ASGM activities. An additional 100 publications with main objectives focusing on the applications of RS and or GIS for the monitoring of formal or informal, small- or large-scale mining activities were also consulted for relevance. The combined results of these various searches account for more than 150 references over the last three decades. This list was

further screened to select only those satisfying the study constraints. In addition, other publications were used to illustrate the concepts, arguments, or examples presented in this article, leading to a total of 67 references used for this work.

*2.2. Assessment of RS Methods Applied to ASGM Sector*

2.2.1. Overview

ASGM activity has different forms of impacts on the environment including air and water pollution, land degradation, as well as land use and land cover changes. All of these impacts can be detected by RS tools and techniques [29,30]. RS tools and techniques are mostly applied to monitor three major variables in the ASGM context:

- The evaluation of deforestation or land cover change caused by the mining processes (generally related to alluvial mines and open pit mines) [31].
- The evaluation of water pollution caused by the mining activity in proximity to rivers or on river channels by detecting water turbidity levels in stream channels.
- Detecting and estimating mercury presence using spectral signatures and assay laboratory confirmations.

2.2.2. Utilized Platforms and Sensors

Optical RS data are obtained from sensor systems mounted on platforms, such as satellites, to detect solar radiation that is reflected from targets on the Earth's surface. Data for optical image processing are commonly available at the top of the atmosphere (TOA) level or as surface reflectance after considering atmospheric influences. As the availability of surface reflectance data varies depending on location and date, the end user may need to carry out atmospheric correction. Depending on the methodology used for image analysis, certain indices have been designed for a specific processing level or perform better at certain processing levels [32,33].

There are several sources of primary RS data. Image-based primary sources of data include multi-spectral satellites such as the United States Geological Survey (USGS) Landsat sensors, European Space Agency (ESA) Sentinel sensors, SPOT-2, CBERS-4, QuickBird, the Japanese Aeronautics Exploration Agency (JAXA), Digital Earth Africa, and the Global Earth Observation System of Systems (GEOSS). Individual scenes can be downloaded from various hubs, e.g., the Copernicus hub (https://scihub.copernicus.eu/, accessed on 10 February 2022) for Sentinel-1 and Sentinel-2 and USGS's Earth Explorer platform (https://earthexplorer.usgs.gov/, accessed on 10 February 2022) for Landsat data and Sentinel-2 worldwide. Copernicus and/or USGS data are also available in hubs such as the Theia data and service center (https://catalogue.theia-land.fr/, accessed on 10 February 2022), Terrascope (https://terrascope.be/, accessed on 10 February 2022), and INPE—Instituto Nacional de Pesquisas Espaciais (http://www.dgi.inpe.br/, accessed on 10 February 2022). To facilitate access to the data, platforms such as AppEEARS (https://lpdaac.usgs.gov/tools/appeears/, accessed on 10 February 2022) allow easier access to analysis ready data while R programming packages and python libraries facilitate the access to the API of various hub; e.g., getSpatialData (https://github.com/16EA-GLE/getSpatialData, accessed on 10 February 2022), sentinelloader (https://github.com/flaviostutz/sentinelloader, accessed on 10 February 2022), and Sentinelsat (https://sentinelsat.readthedocs.io/en/master/install.html, accessed on 10 February 2022).

2.2.3. Approaches and Tools for Data Analysis

Optical imagery can be analyzed using diverse approaches such as image classification, image transformation using indices, and feature targeting approaches. Image classification models in RS applications sometimes use machine-learning (ML) algorithms, which are suitable for modeling complex class signatures, accept a variety of input predictor data, and do not necessarily require knowledge of the data distribution (i.e., are

non-parametric) [34]. ML algorithms can operate supervised and unsupervised learning with the former requiring labelled training data while the latter operate through clustering and association techniques [35]. For supervised learning the user must feed the model with interpreted (i.e., labelled) training data. The ML algorithms that are most frequently used in multispectral image-classification are the random forest (RF) classifier, support vector machines (SVMs) (supervised), decision trees (DT), and artificial neural networks (ANN) [31]. It is important to note that there is no stand alone, one-size-fits-all methodology for image classification. The choice of techniques is contingent upon but not limited to: (1) the objective of the study, (2) image data accessibility for the area of interest and objectives, and (3) availability of and access to relevant image processing software.

Where atmospheric conditions (e.g., cloud cover and seasonal burning) disturb the use of even high-resolution satellite imagery to observe ASGM areas, the use of radar may be able to overcome the challenges of optical satellite imagery. Radar can penetrate cloud-cover to detect pit subsidence as well as land use and land cover changes. Nevertheless, high precipitations have been proven to affect the accuracy of satellite data [21]. Radar interferometry uses multiple radar images of the same area, which have been taken on different dates and times for change detection. Historical images show the host region and its existing conditions before the emergence of ASGM, while current scenes reveal the growth of ASGM and its related effects. In this regard, two general approaches can be used. These are: (1) InSAR, which typically uses succeeding radar images to increase the information in a scene or to develop a digital elevation model (DEM), and (2) repeat-pass interferometry, which also uses radar scenes of the same area but on different passes of the satellite.

The main software and tools for the processing of spaceborne imagery and spectral geospatial data include ArcGIS, ERDAS IMAGINE, ENVI, ILWIS, IDRISI, Orfeo ToolBox (OTB), SNAP, Multispec, and QGIS. Among these, OTB, SNAP, Multispec, and QGIS are free software. Furthermore, open-source packages in R [36] and Python are available [16]. As datasets can be large, especially in the case of time-series analysis, cloud computing has become a critical component of RS data storage especially in the case of wide area and long-term monitoring of ASGM activities. Various solutions using Python, R, and Javascript APIs are available and include Google Earth Engine, the Open Data Cube, OpenEO, and SentinelHub, all of which have their benefits and constraints [37].

### 2.2.4. RS for Deforestation and Landcover Change

Pixel-based and object-based approaches have greatly contributed to studying ASGM activities and changes on land. The most common approach is pixel-based, although this sometimes comes with a new form of challenge regarding misclassification in mapping mining-related landcover [24,38]. Approaches utilizing pixel-based classification followed by post-classification improvements through the appropriate knowledge of the mining setting have been successful in overcoming these limitations and reducing omission and commission errors [16,39]. It is important to take note of periods within a given year where the differences between mining sites and bare soils are more evident and unambiguous. An example is the dry season [40]. Medium resolution multispectral imagery has also been used in combination with nighttime light and precipitation data to identify the emergence of expansions in built-up remote areas. This can serve as a proxy for dwellings of miners, indicating the presence of ASGM activities in a particular locality [41]. Combinations of both high resolution and medium resolution data have been used to also improve the detection and details of specific areas of interest [42,43].

Recent studies on deep learning (DL) techniques (i.e., neural network algorithms involving a higher number of hidden layers) suggest that the convolutional neural network (CNN) is a valid candidate for land cover classification purposes and can outperform those with omission and commission errors as low as 8% in the context of ASGM mapping [44]. However, the literature showing how CNN should be applied for land cover tasks related to ASGM is still limited [44].

On the other hand, object-based image classification comprises two procedures, namely, segmentation and classification, usually performed on high resolution images [45]. In image segmentation, image objects are delineated based on the homogeneity of pixels and spatial contingencies, continuous and contiguous objects [38], overcoming the limitations of pixel-based approaches. Yet, errors in the segmentation process could highly reduce the quality of the output [46]. A subset of these methods, used primarily in deforestation studies, can detect sub-pixel changes, which eventually reduces the problems caused by spectral mixture analysis [47–49].

Classification models do not necessarily need the total spectral information. An effective band/feature selection process would result in enhanced performance of the model in terms of costs and accuracy of the results [45]. For instance, it has been shown in a case study in Ghana that the Sentinel-2 Band 5 (band center 705 nm) was the highest contributor to a land cover classification and, more importantly, it contributed most to delineating mining sites [19]. Classification models can also use multi-band indexes as input data such as the normalized difference vegetation index (NDVI) [13,19,31]. However, note that NDVI is influenced by many environmental factors such as topography, bare soil conditions, atmospheric conditions, vegetation association, rainfall, and non-photosynthetic materials [50]. Other indexes such as the soil-adjusted vegetation index (SAVI), the modified soil-adjusted vegetation index (MSAVI) and the transformed soil-adjusted vegetation index (TSAVI) are used to feed the classification models with enhanced performance especially in low vegetation areas [31]. Finally, NDVI could also be used in post-classification steps to reduce the uncertainty over land cover changes after the determination of a threshold that separates seasonal change influence from artificial influence on land cover change. The images produced using classification methods can eventually be used to determine where land cover has changed over time and to calculate areas that have evolved into artisanal mining sites. This can be achieved by performing a change detection process over pairs of images and determining the evolution of land cover between two time points.

The approach is slightly different in the alluvial wetland setting where deforestation is less of a concern than wetland and riparian area destruction and its impacts on water quality. As the multispectral signature of bare soil, especially river-bank sediment, and open mines are relatively similar, the detection of mining sites can be challenging [31]. Furthermore, such a setting is very challenging for cloud and shadow detection in satellite imagery, thus leading to potential errors in the mapping of the mining sites [16]. One approach applied in this context is morphological profiling, which is run on the output of the classification model to differentiate mining sites from bare soil. The success of the morphological profile can be attributed to its ability to isolate bright and dark structures in images, and by exploring a range of different spatial domains as well as brightness and darkness contrast [31]. This approach facilitates distinguishing non-vegetated areas that could have been identified quite simply with a classifier that is relatively insensitive to the illumination and albedo effects that are common in rugged terrains [31]. The spectral angle mapper (SAM) is another approach that enables high quality detection of cloud and shadows in the context of ASGM analysis [18]. A third method involved is the classification of landcover followed by a post-processing step to distinguish mining sites and mining ponds due to the required condition of their simultaneous occurrences [39].

The feasibility of the intermittent small baseline subset (ISBAS) interferometric synthetic aperture radar (InSAR) method together with Sentinel-1 imagery for monitoring ASGM activities has been explored by Ding, et al. [50]. The study found a high level of subsidence based on surface motion values, which is a clear indicator of mining activity. Other indicators include surface deformation, bare soils, and water pollution. Several simulation results show that the European Space Agency Copernicus Sentinel-1A/B constellation is capable of mapping rapid ASGM activities in the landscape. For instance, Forkuor, et al. [21] used annual time-series Sentinel-1 data to map and monitor ASGM activities along major rivers in south-western Ghana. A change detection approach based

on three time-series features was used to compute a backscatter threshold value suitable for detecting mining-induced land cover changes.

### 2.2.5. RS for Detecting Impact on Rivers

The impacts on rivers can be either due to the direct dredging activities in the water channel or due to runoff from land excavations. Detection of dredging in rivers is challenging, yet can be achieved either through detecting a plume of sediments using medium resolution imagery [43] or through detecting the dredging vehicles using high resolution data [46]. Various bands and indexes have proven useful in detecting water turbidity and suspended sediment and can be used to identify the impacts of ASGM along rivers, especially, when considering long time series of satellite data [17,51]. Examples include the modified normalized difference water index (MNDWI), the Band 8A-VRE 4 and the Band 3 (Green) in Sentinel-2-A data [19]. Another example is the combination of Landsat 8 bands; (4,3,2) to distinguish deep water from shallow water, (5,6,4) to distinguish water from ground and, (6,5,2) to distinguish bare ground from ponds [43]. The effects of precipitation on water turbidity, and on the values of the index that is used to infer it, can be minimized by determining the threshold between naturally and human-induced turbidity [43]. In the case studies, the MNDWI values were collected at different times within the dry season, at specific locations along a river in proximity to a known ASGM site to identify the effects of ASGM activities. Collected values could be analyzed using unsupervised techniques or using manual approaches [43]. Features with similar spectral properties that are potentially misinterpreted by the classification model can be corrected in the post-classification stage with different methods handling manual to automated operations [21].

### 2.2.6. RS Supporting Mercury Presence Estimation

RS data can be integrated with environmental data to better analyze the influence of the mining process on the biota/mercury content in the environment. For instance, the location of mining sites from existing datasets or from field work can be utilized to create reference data that train supervised classification models or to refine model results with the assumption that a non-existing mine in recent high-resolution images implies that there was no mine present in the same location even in the past [19,44,52–54]. Similarly, the turbidity of water may be used as a proxy for mercury content in water if the RS data are combined with in-situ water sample data that are taken on specific dates that correspond with the available RS data [17,20]. In a case study in Colombia by UNODC [43], additional data were derived from the results of the RS approaches such as: (1) the direction of expansion of the mining sites through time, (2) the amount of people being affected by the polluted waters resulting from the mining activity, and (3) the coexistence of illegal cultivations and ASGM sites. This could be achieved by: (1) analyzing time-series data on mines size and location, and (2) integrating external GIS data such as the delimitation of watersheds, the gridded population data, and the location of illegal cultivation spots. The information obtained could be used to orient government policies and actions towards specific directions that deserve the highest priority.

## 3. Case studies Using the Open Data Cube and MapX

### 3.1. Overview

Two case studies are considered to illustrate the opportunities provided by satellite data for the detection of ASGM and its impacts. They make use of the Data Cube on Demand (DCoD) for data access and processing, and MapX for data sharing. DCoD [55] is an Earth observations Data Cube (EODC) concept that facilitates the generation and use of an EODC instance virtually anywhere in the world. It requires the user to specify an area of interest, select the spaceborne sensors of which data is required, and choose a desired temporal window. For the case studies, Sentinel-2 scenes from Google Cloud were indexed in the DCoD using Python and R scripts. The attained results were published

using MapX; an online open-source platform for mapping and visualizing geospatial data on natural resources [56]. MapX has been developed by UNEP and UNEP/GRID-Geneva (https://unepgrid.ch, accessed on 10 February 2022), and is anchored as a key component of the World Environment Situation Room, which is the UNEP data, information and knowledge platform.

### 3.2. *The Case of Land Cover/Land Use Monitoring*

This case study considers two study areas located in the province of South Kivu in eastern Democratic Republic of the Congo. The first focuses on the Bipasi and Kazibe mines located in the western part of the mining town of Kamituga, while the second covers a wider area including Kamituga as well as the region to the south where several dozen mines are located [57].

In this case study, land cover changes were monitored using the vegetation fractional cover (VFC) [58] that estimates the fractions of photosynthetic vegetation (PV), non-photosynthetic vegetation (NPV), and bare soil (BS) for each pixel. Although originally developed for Landsat 5/Landsat 7 products, Sentinel-2 products were used as their 10m resolution is more suitable for monitoring artisanal mining activities than Landsat products (30m resolution). A first visual inspection has been conducted to assess the performance of the VFC classification. A first check was made using reference satellite images combined with the location of the mines visited by International Peace Information Service (IPIS). IPIS has previously compiled various maps and dashboards on ASGM in the area [59]. This assessment showed a good correspondence between the model and the observations where it was realized that non-photosynthetic vegetation corresponds to the mining area, and bare soil corresponds to the built-up areas consisting of towns and villages appeared. For each of the study areas, a layer with a dashboard (https://app.mapx.org?project=MX-IY9-QCF-ILZ-UVO-07Y&views=MX-BD2ZB-CPRZ6-ISSWP,MX-QSNYV-VWM4T-1T4NT,MX-RQ6YP-SP29M-Z01X6&lat=-3.899&lng=20.376&z=5.256&viewsListFlatMode=true&language=en, accessed on 2 June 2022) was developed in MapX to visualize land cover changes in an interactive and comprehensive way (Figure 1).

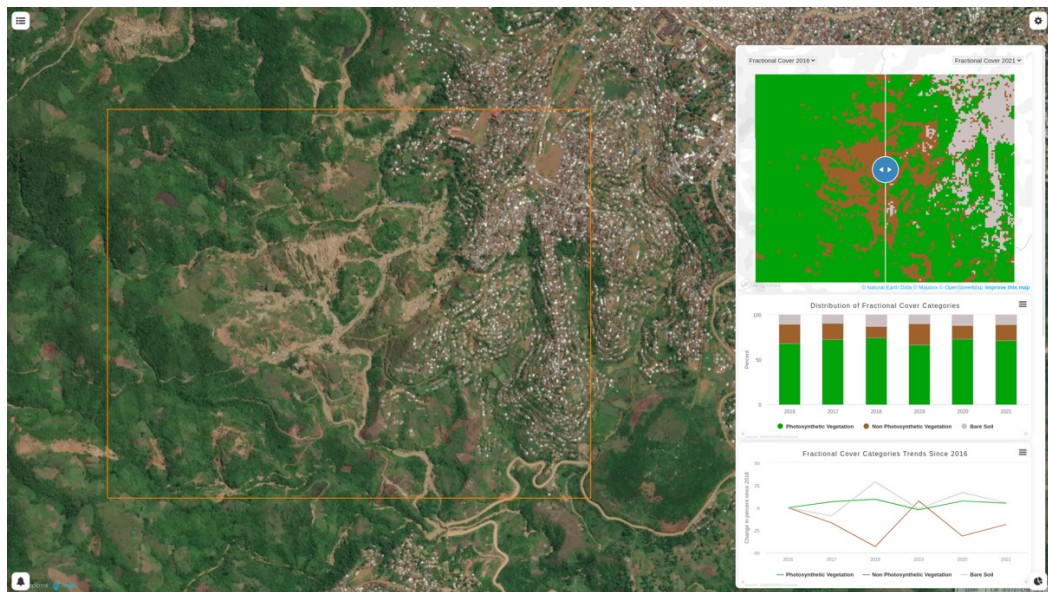

**Figure 1.** Screenshot of the MapX layer developed for the study area covering the Bipasi and Kazibe mines.

Although the VFC algorithm was not developed specifically to monitor ASGM activities, it produced results that enable the quantification of the evolution of surfaces

exploited by the mines, the gain or loss of vegetation, and urban growth. Based on a first visual inspection of the results, the classification is robust when the study area is defined at site level. At this level, the land cover is more homogeneous, and the findings confirm the hypothesis that NPV is a suitable tool for analyzing mining areas. Calculating VFC at the larger scale in heterogeneous landscapes introduces misclassification of some fields and river sections that are rich in alluvium. This decreases confidence in the results.

### 3.3. The Case of Water Turbidity

Here, a case study was carried out in two study areas to monitor the impact that ASGM could have on river water quality. The first area was centered on the mine of Mambo Bado south of Nia-Nia, a small town located in the Orientale Province in the Democratic Republic of the Congo, and the second covers the city of El Bagre, in the department of Antioquia in Colombia [39]. In the analysis, total suspended matter (TSM) was investigated in the DCoD and used as a proxy to monitor water turbidity in rivers where artisanal mines are located upstream [60,61]. Sentinel-2 scenes were indexed in the DCoD for the two study areas. Before calculating the TSM, an algorithm was used to classify water. Then a filter was applied to keep only pixels with a water content greater than or equal to 50% thus retaining only rivers and water bodies for the calculation of TSM [62]. The quality of the results depends on the cloud cover, the width of the river, and the vegetation along the river (Figure 2). For the two study areas, some results have been discarded as their quality was not sufficient to monitor suspended matter. Then, a visual inspection of the results showed that the mines in both areas could be contributing to a local increase in TSM. Such preliminary observations are encouraging enough to explore further case studies and conduct a comprehensive accuracy assessment.

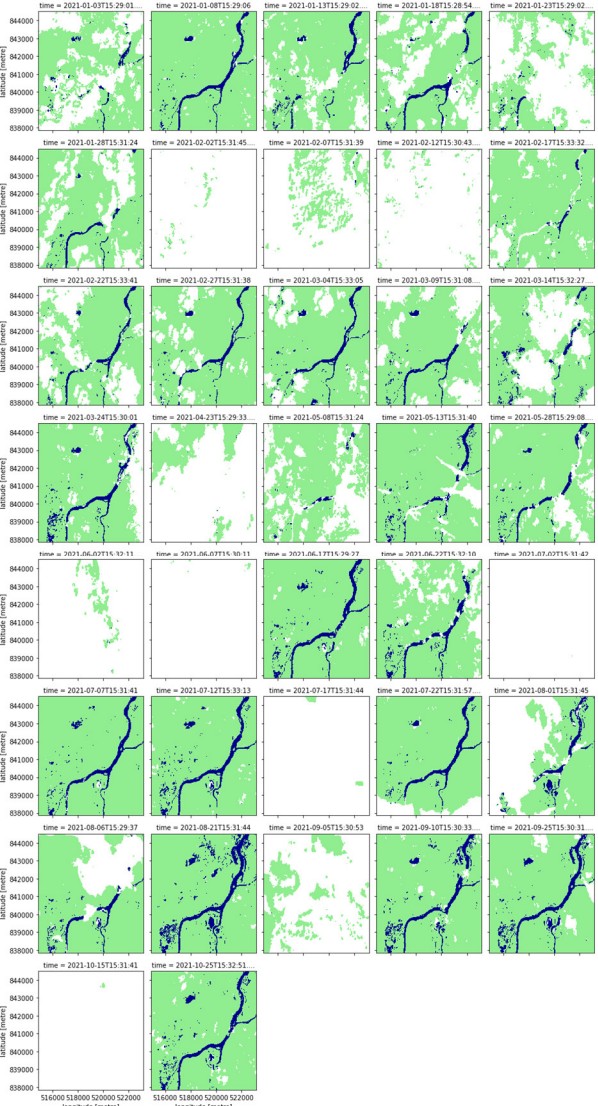

**Figure 2.** Water classification results over the period 1 January 2021/31 October 2021 around the town of El Bagre showing variability in Sentinel-2 image quality. Results are classified as follows: blue = water; green = not water, and white = pixel removed by the clean mask for the analysis.

A layer (https://app.mapx.org?project=MX-IY9-QCF-ILZ-UVO-07Y&views=MX-BD2ZB-CPRZ6-ISSWP,MX-PN6KV-LIFUJ-BSO8R,MX-XTX2N-M6GRE-QZZB0&viewsListFlatMode=true&language=en&, accessed on 12 May 2022) was developed in MapX for each study area to visualize suspended matter changes over time using a slider (Figure 3).

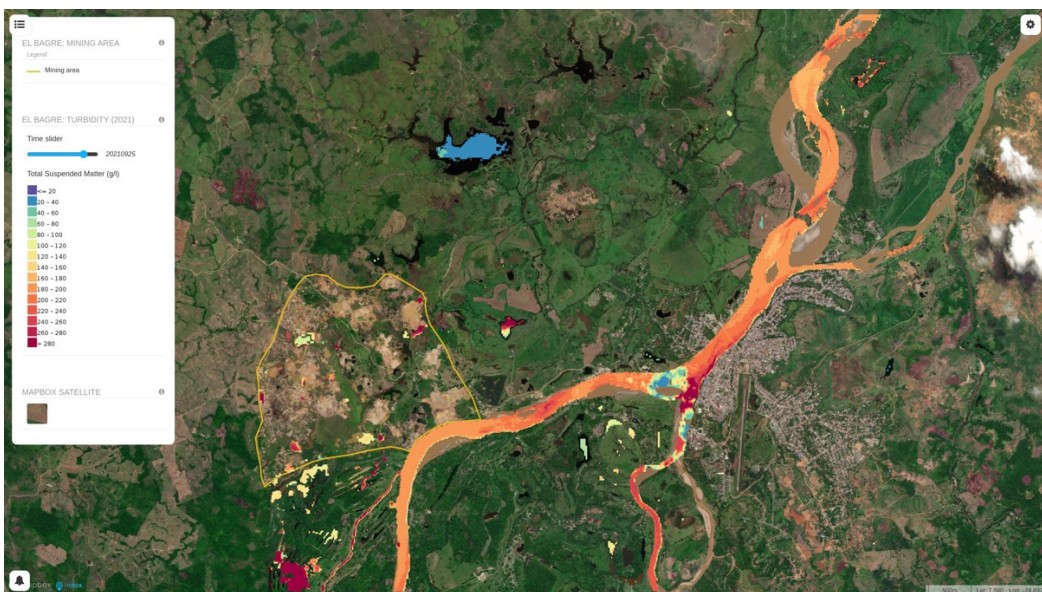

**Figure 3.** Screenshot of the MapX layer developed for the study area covering the city of El Bagre overlaid on high resolution satellite imagery.

## 4. Discussion

### 4.1. Observations

The use of satellite imagery for mapping changes attributed to ASGM activities and as an indication of the use of mercury by ASGM practitioners have been demonstrated in the series of case studies conducted as part of this study. According to the UN (95th Plenary meeting, 3 December 1986), the general purpose of Earth observation and RS is to improve natural resource management, environmental protection, and land use. The positive results documented in the literature and the case studies presented in this article show that a variety of RS techniques can be used for ASGM monitoring depending on the objectives, on data availability, and on the geographical and morphological contexts. In particular, the technology can help determine spatial dynamics (everything mappable on the Earth's surface) and biogeochemical parameters like mercury concentrations and turbidity in biotic and abiotic matrixes. If field data collection is planned and performed in a coordinated manner, it becomes easier to connect the three parameter classes.

Given the nature of ASGM, governments require extensive data resources and analysis to monitor, implement, and enforce laws and policies. Recently, the availability of satellite imagery has been increasing not only in the number of images, the frequency of available images per region, and image resolution, but also in terms of platforms that offer pre-processed, cloud-free data that is analysis-ready. These include Google Earth Engine (GEE) [63], Microsoft Planetary Computer, Food and Agriculture Organization, and SE-PAL [64]. The increasing availability of platforms that offer ready-to-use RS data and server-side calculation power could facilitate the use of RS techniques for ASGM monitoring in application projects. They could also promote the development of replicable methodologies which would produce more harmonized information/indicators at different geographical scales and promote the use of RS techniques for policy making.

Techniques such as photography from unmanned aerial vehicles (e.g., drones) and sediment, vegetation, and water samples can be used additionally to train image classification models. Statistical data resulting from these classifications can be used to assess the presence and levels of use of mercury for gold production in a locality. For instance, in one of the case studies presented in this article, fieldwork was conducted in Kamituga in eastern Democratic Republic of the Congo by the IPIS. This fieldwork included the collection of geologic and geomorphic data through measurement, observation, and sampling of ASGM sites. Sampled sites were then studied using satellite imagery to identify the

extent of environmental changes. The use of unmanned aerial system (UAS) imagery (such as those captured by drones) to map alluvial deposits in ASGM regions has also been explored in recent scientific studies [65]. However, these methods require significant expertise on the ground to handle the equipment before meaningful knowledge can be acquired. RS techniques allow a detailed mapping and monitoring of ASGM activities and the development of high-resolution geomorphic models for identifying resource deposits. A combination of these technologies can enhance the capacity for rapid assessment and mapping of environmental, social, and economic impacts of the ASGM activities. The diagram proposed below summarizes how RS techniques can be combined with in situ inputs to monitor ASGM (Figure 4).

**Figure 4.** Proposed methodology for ASGM monitoring.

Important end goals could be the focus of the analysis, such as protected areas, critical ecosystems, and vulnerable populations, as demonstrated in the case study in DR Congo. This can make it easier for governments to demarcate concessions for miners in the formalization processes of their activities. In the previous studies, few have considered the relevance of such algorithms for harnessing the opportunities of RS towards addressing the critical global challenge of ASGM.

Study Limitations and Prospects

As explained by Hintjens [2], mining issues are location-specific and cannot be generalized. This study had limited access to all developing countries specific issues with regard to ASGM. Therefore, this study dwells on general issues common to most

countries. However, applications of the RS techniques are adaptive and can address most local and context-specific monitoring issues. Meanwhile, more case studies should be conducted across all countries hosting more than significant ASGM activities [3]. This would generate relevant data for the detection of nuanced biophysical and spectral differences, which would further help in data calibration and signature segmentation for the varied environments. With this data, the applications of RS would provide robust information to address the location-specific needs of every locality with its unique biophysical characteristics. To achieve this requires dedicated funding from local government agencies and relevant ministries. Thus, funding is required for the building of robust spectral libraries for regions with similar biophysical features. However, a foreseen challenge in this regard is data storage and protection. To address this would require investments in cloud computing storage facilities. Hence, one limitation identified in this study is the capacity of developing countries to store and manage large volumes of RS data at the same time. Cloud storage is an ideal solution but with the cost involved, different data management systems for the storage of RS data to address developing countries specific needs should be explored. Also, the study could not explore the capabilities of RS for linking the impacts of ASGM to socio-economic activities of local areas directly. More studies are required to demonstrate how RS can be used to quantify the direct socio-economic impacts of ASGM rather than using proximate analysis.

### 4.2. Recommendations

Governments are encouraged to adopt RS methods into their ASGM monitoring plans and policies, including national action plans for ASGM under the Minamata Convention on Mercury [66]. The most appropriate techniques should be selected, however, based on contextual factors such as: (1) the objective of the study, (2) image data accessibility for the area of interest, and (3) availability of and access to relevant image processing software. The integration of different RS techniques with field data in the monitoring process is encouraged as it can increase the reliability of the results. Specifically, the integration of in-situ measurements, Indigenous knowledge, and socio-economic data helps to provide better quality data, improve understanding of causal relationships between different factors, and could also be used for training and validating the outputs of ML/DL algorithms.

The broad availability of free satellite data and platforms offering ready-to-use data and processing capabilities should be seen as an opportunity for governments to adopt RS techniques in their ASGM monitoring and policy strategies [67]. Similarly, the same should encourage software developers to continue offering these services and to facilitate, where possible, their use through well-designed interfaces for end users. Since RS techniques require technical competences, public sector staff such as the responsible personnel of mining regulatory agencies are encouraged to: (1) build their capacities and competence in the manipulation and operations of RS software, and (2) collaborate with the scientific community and local universities for such capacity-building endeavors and consultations. International organizations and funding agencies are encouraged to engage in discussions and dissemination of the opportunities of RS. Similarly, international organizations are encouraged to facilitate the use of RS on monitoring ASGM activities in remote areas. Capacity-building of local stakeholders (researchers, government officials, local communities' representatives, artisanal miners) must be harmonized and enhanced to facilitate a participatory and partnership approach with local communities and Indigenous peoples in ASGM monitoring policies.

An integrated use of high-resolution optical imagery, current and historical aerial photographs, SAR images, and DSMs could be helpful in situations where mines are not easily distinguishable from the surrounding areas. Image classification models of the RS data should ideally be fed with training data consisting of a large set of labelled data based on: (1) existing and historical geographical datasets of mine extents at a given time; (2) existing and historical water, air, soil, and vegetative cover data; (3) collaborative

mapping of artisanal mine sites on high resolution true-color images by local people; and (4) evidence from fieldwork such as soil, vegetative, and water samples, as well as notes, pictures, videos, and local knowledge. Post-classification imagery processing can improve the distinction of classes that are difficult to separate in certain environments such as bare soil and mines in dry land.

Location association is a spatial analysis technique which can estimate the amounts of water or soil mercury contamination with respect to the distance from an identified ASGM site. This technique is based on the hypothesis that the existence of ASGM activities in a particular place may be an indicator of the presence of mercury in nearby waterbodies, soils, food crops, and plants. If seasonality has a role in land cover changes in the area throughout the selected series of images, its role could be quantified with statistical methods using the RS data after the classification. In case of dry environments, a morphological analysis could be helpful in further separating bare soil from mining sites. This might be useful in the case of hard rock mines as they are likely to feature a depressed morphology compared to the surroundings, but it would not make a significant difference when looking for alluvial mining sites which are generally located along rivers.

Laboratory analysis of field samples can help us understand the physical and chemical properties, including contamination with mercury. This can help to determine: (1) spectral reflectance and signatures in a given geological area; (2) the extents of surface, subsurface, and structural contamination in the environment; (3) estimates of the associated potential health and/or environmental impacts; and (4) decisions on mitigation, remediation, and reclamation measures needed at ASGM sites. This process requires a basic knowledge of the geology of the area (rocks and soils) and an analysis of the mercury concentration, healthy vegetation, healthy soils, and clear water (indices such as NDVI, NDWI, MNDWI, and SAVI are mostly used to aid analysis and understanding).

Using historical data, it is possible to identify areas which have been mined for a long time or that were previously mined and abandoned. Such places with a long history of ASGM operations can be mapped as potential hotspots due to the accumulation of mercury in the soils and the presence of both underground and surface water reservoirs. This is an important mechanism for detecting hotspots and building mitigation and restoration algorithms. This requires linking RS data with field sampling as described in objectives two and three above. A reverse analysis of historical RS data baseline conditions of existing and previous sites would facilitate linking the spectral signatures of samples from trees and shrubs to satellite data as mercury contamination may produce a unique coloration in the spectra.

There is a unique opportunity for regulatory agencies to directly collaborate with RS data providers for real-time monitoring of ASGM activities to facilitate the adoption of the proposed algorithms. This may be done, for example, by setting up sub-stations across ASGM zones to transmit high resolution satellite data to the main receiver station. These data should be processed on the spot for: (1) mercury hazard identification, (2) monitoring the emergence of new activities and/or expansions of existing activities and spillage beyond standard thresholds, and (3) real-time feedback mechanism to regulators, Indigenous peoples and local communities, and miners. To sustain the robustness of real-time monitoring algorithms, it is important to train local regulatory operations and supervisory teams in ASGM areas on the use of GPS and mechanized mobile phones for prompt reporting to sub-stations and onward transmission to the main station. Monitoring based on real-time RS data can also serve as early warning systems to support governments addressing the observed changes and preventing the harmful effects of mercury on health and the environment.

## 5. Conclusions

This work assesses the use of RS technologies for monitoring ASGM activities in developing countries. RS techniques are a valuable means to provide consistent information on ASGM activities and provide complementary quantitative data to field measurements

to support national priorities such as policy making and implementation, public health interventions, baseline setting, and monitoring. It also helps to have harmonized information/indicators at different geographical scales. Following the algorithms introduced in this paper, policymakers can efficiently and effectively support monitoring activities and further policy developments.

From a research perspective, efforts should be directed towards improving land use/land cover methodologies applied to ASGM. In particular, the use of ML/DL techniques together with data fusion techniques (e.g., optical, radar, UAV, lidar, in-situ, crowded-sourced), time-series analysis and stack of analysis ready data organized in Data Cubes are relevant means to reliable and consistent land use/land cover information.

Finally, international organizations, and donors and other relevant stakeholders are encouraged to build on and integrate these algorithms into projects and monitoring programs for policy development, implementation, and evaluation in the ASGM sector. This approach is recognized to be particularly useful, especially in remote areas where ASGM may be widespread but difficult to detect. Collaborative efforts with RS scientists through ASGM monitoring projects would facilitate development and refinement of the methodologies to provide relevant information to support policy making and implementation.

**Author Contributions:** Conceptualization, G.G., K.D., M.M. and P.L.; methodology, G.G. and M.P.; software, T.P.; validation, E.I., K.D. and M.M.; data curation, T.P.; writing—original draft preparation, A.B., A.-W.M., E.I., G.G., M.P., P.L. and T.P.; writing—review and editing, A.-W.M., E.I., G.G. and P.L.; visualization, A.-W.M. and T.P.; supervision, G.G. and P.L.; project administration, K.D., M.M. and P.L.; funding acquisition, K.D., M.M. and P.L. All authors have read and agreed to the published version of the manuscript.

**Funding:** This research was funded by the Norwegian Environmental Agency, grant number SB-011030.03.01.04.

**Data Availability Statement:** The three geospatial layers developed in the frame of the case study using the Open Data Cube technology for monitoring land cover changes can be accessed from: https://app.mapx.org?project=MX-IY9-QCF-ILZ-UVO-07Y&views=MX-BD2ZB-CPRZ6-ISSWP,MX-QSNYV-VWM4T-1T4NT,MX-RQ6YP-SP29M-Z01X6&lat=-3.899&lng=20.376&z=5.256&viewsListFlatMode=true&language=en, accessed on 2 June 2022. The results of the two case studies for monitoring water turbidity in DR Congo and Colombia (using the Data Cube technology) can be seen from: https://app.mapx.org?project=MX-IY9-QCF-ILZ-UVO-07Y&views=MX-BD2ZB-CPRZ6-ISSWP,MX-1JRWJ-DGN9M-YR9IR,MX-PN6KV-LIFUJ-BSO8R,MX-XTX2N-M6GRE-QZZB0&viewsListFlatMode=true&language=en& (accessed on 2 June 2022).

**Acknowledgments:** We would like to acknowledge the Norwegian Environment Agency for funding this research. We also thank the Minamata Convention Secretariat and UNEP for their valuable inputs and International Peace Information Service (IPIS) for their review of the D.R. Congo case study. Finally, we express our gratitude to the reviewers of our research for their valuable comments: Karin Allenbach (UNEP/GRID-Geneva), Eisaku Toda (UNEP), Birane Niane (University Iba Der Thiam, Thiès, Senegal) and Robert Moritz (University of Geneva, Switzerland).

**Conflicts of Interest:** The authors declare no conflict of interest. The funders had no role in the design of the study; in the collection, analyzes, or interpretation of data; in the writing of the manuscript, or in the decision to publish the results.

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
