# Peer review of "Assessing the Applications of Earth Observation Data for Monitoring Artisanal and Small-Scale Gold Mining (ASGM) in Developing Countries"

_remotesensing, doi:10.3390/rs14132971_

Round 1

Reviewer 1 Report

The paper is now ok.

Author Response

We thanks the reviewer for his valuable comments during the review process.

Reviewer 2 Report

Dear Authors,

The presented manuscript was correctly edited and written with satisfactory academic language. The research results obtained are supported by the methodology presented. The quality of the graphics is correct. The considerable amount of cited literature indicates that the authors are well prepared in current state-of-the-art.

In my opinion, the article can be published in its current form in Remote Sensing.

Congratulations!

Reviewer 

Author Response

(The authors gave the same response as above.)

Reviewer 3 Report

Dear authors:

The research topic of this article is valuable and the research content fits the journal topic. On the whole, the article is well completed, the logic of the article is clear, and the author has done a lot of research. The authors provide a large number of data sources and provide a detailed analysis of the application of RS in several aspects, while introducing scientifically detailed methods. At the same time, this article cites cases, and elaborates how RS is applied in practical cases, which enriches the practical value of the article. Suggest a little revision:

1 In the first part of the text, the marginal contribution of this paper can be expressed more clearly

2 Can the author add a little more content in the Discussion section to explain the research limitations of this paper and future research directions.

3  Figure 4 is not very clear, can you increase the resolution of the picture?

Best regards!

Author Response

Comment: 1 In the first part of the text, the marginal contribution of this paper can be expressed more clearly.

Response: Thanks to the learned reviewer for the objective observation. The marginal contributions of the paper have been revised for clarity of expressions to the wider readership of the paper should it be published. Please, find these revisions in lines 68-69, 88, and 95 page 2; and lines 101-116, page 3.

Comment: Can the author add a little more content in the Discussion section to explain the research limitations of this paper and future research directions

Response: Many thanks to the learned reviewer for the advice. In this revised edition of the manuscript, a new third level subsection has been inserted and titled; 4.1.1 “Study Limitation and Prospects”. This subsection discusses the study’s limitations and recommends some future directions that could help to address the limitations identified in this paper.

Please, find these revisions on page 13, lines 471-490.

Comments: Figure 4 is not very clear; can you increase the resolution of the picture?

Best regards!

Response: Thank you for the close observation. The resolution of Figure 4 has been improved and has been updated in this revised manuscript. Please, find the updates on page 12.

This manuscript is a resubmission of an earlier submission. The following is a list of the peer review reports and author responses from that submission.

Round 1

Reviewer 1 Report

Authors tend to present a framework for monitoring artisanal and small-scale gold mining using earth observation data. And comments are listed as follows.

1 Wring of this manuscript should be improved. This manuscript is like a report rather a scholar paper.

2 In title, abstract and introduction, authors mention that a framework is proposed in this manuscript. However, in the following part of this manuscript, no more description on framework is given.

3 Areas of artisanal and small-scale gold mining are not demonstrated in any results in this manuscript. What’s the relationship between gold mining monitoring and figures in this manuscript?

4 Novelty and contributions of this manuscript should be strengthen and highlighted.

5 Quantitative analysis should also be presented.

Author Response

Please find our answers in the attached PDF document.

Reviewer 2 Report

The article presents a review on remote sensing to monitor Artisanal and Small-scale Gold Mining (ASGM) activities, presents two study cases of Remote Sensing (RS) to monitor ASGM, and protocols for monitoring AGSM sites.
The theme is significant and relevant for the RS community and policymakers in different dimensions, such as these activities' economic and environmental aspects. 
In section 2.2.2 the authors list primary sources of EO data used in studies about ASGM activities. They include a list of organizations (e.g., USGS), cloud platforms (e.g., Google Earth interface), and satellite/sensors (e.g., QuickBird). The platform  CBERS-2 reached its end of life, while there are two CBERS flying: CBERS-4 and CBERS 04A, and another platform Amazonia-1. A suggestion to the authors is to update and better organize this list and specifically cite INPE as an agency. 
The systematic review is attractive and comprehensive; however, it could be more concise regarding the fundamental aspects of RS (image processing methods, for example); thus, in my opinion, references 46 and 52 to 56 are too generic and irrelevant to the article's main topic.
"Unfortunately, the literature showing how the CNN should be applied for land cover tasks is still limited" I disagree with this statement unless it refers specifically to land cover related to ASGM; in this case, it should be explicit.
In the case studies is not clear how the "illicit" aspect of the ASGM activities was characterized (line 548).
The presented protocols are reasonably simplistic and should be read more like guidelines and presented as such (e.g., 4.2.5).
Finally, the text needs reviews: there are two Figures 1 and misspelled words such as "Resultats" "whit e= pixel"  "themdevelop" .

Author Response

(The authors gave the same response as above.)

Reviewer 3 Report

The title indicates that the authors propose a framework, but this term has only three occurrences in the text: one in the title, one in the abstract, and one in the introduction. In the document, the authors propose a protocol, not a framework. However, the protocol is described only in the reccomendations subsection within discussion section. It is expected that the main contribution of a manuscript (as stated in the title) should be proposed not as a reccomendation, but early in the text in a way that the experiments are developed to test whether the ideas related to the proposed contribution are feasible.

The protocol (figure 4) mixes land cover classes, methodologies, and data. It has arrows to nowhere (below Space Data), and arrows with different widths, but the authors do not point out the underlying semantics for such widths. There is also no direct relation between the boxes of the figure ("Proposed RS protocol") and the title of the following subsubsections in 4.2 ("Proposed RS protocols"). Finally, there is no explicit connection between the literature/experiments and the proposed protocol.

The methodology section focuses only on the literature review. The methodologies of the experiments are described in the same section of the results (Section 3, Case Studies). Additionally, using visual analysis (lines and 416 and 451) is not enough to support the claim that the results are good. It is necessary to present quantitative results (not only maps) over time, as the objective is monitoring.

In Table 1 (l. 509), the authors point out one of the limitations of RS for ASGM as being a "need for high technical competences for processing". However, in the conclusions the authors state that "It offers, simple, replicable, cost-effective, synoptic, scalable and rapid alternative to derive information on ASGM activities". How can it be simple (and rapid) if there is a need for high technical competences?

l. 183: "A final list of 81 publications satisfying the study constrained was used. Aside from the state-of-the art literature search, about 20 other publications were used to illustrate the concepts, arguments or examples presented in this article." However, the references section has only 75 citations.

l. 110: Using Sentinel-1 as an example related to the fact that RS data is available for the last 4 decades is not a good idea, as it was launched less than a decade ago.

Author Response

(The authors gave the same response as above.)
